# Frailty, Complexity, and Priorities in the Use of Advanced Palliative Care Resources in Nursing Homes

**DOI:** 10.3390/medicina57010070

**Published:** 2021-01-14

**Authors:** Emilio Mota-Romero, Beatriz Tallón-Martín, María P. García-Ruiz, Daniel Puente-Fernandez, María P. García-Caro, Rafael Montoya-Juarez

**Affiliations:** 1Primary Care Center Dr. Salvador Caballero García Andalusian, Health Service, Government of Andalusia, 18012 Granada, Andalusia, Spain; emilio.mota.sspa@juntadeandalucia.es; 2Geroinnova Nursing-Home, Fuente Vaqueros, 18340 Granada, Andalusia, Spain; beatriztm312@gmail.com; 3Caridad y Consolación Nursing-Home, 23008 Jaén, Andalusia, Spain; piedadgarciaruiz@gmail.com; 4Doctoral Program in Clinical Medicine and Public Health, University of Granada, 18016 Granada, Andalusia, Spain; 5Department of Nursing, Mind, Brain and Behaviour Research Institute, University of Granada, 18016 Granada, Andalusia, Spain; mpazgc@ugr.es (M.P.G.-C.); rmontoya@ugr.es (R.M.-J.)

**Keywords:** frailty, complexity, prognosis, palliative care, palliative care complexity, nursing homes, patient transfer

## Abstract

*Background and objectives:* This study aimed to determine the frailty, prognosis, complexity, and palliative care complexity of nursing home residents with palliative care needs and define the characteristics of the cases eligible for receiving advanced palliative care according to the resources available at each nursing home. *Materials and Methods:* In this multi-centre, descriptive, and cross-sectional study, trained nurses from eight nursing homes in southern Spain selected 149 residents with palliative care needs. The following instruments were used: the Frail-VIG index, the case complexity index (CCI), the Diagnostic Instrument of Complexity in Palliative Care (IDC-Pal), the palliative prognosis index, the Barthel index (dependency), Pfeiffer’s test (cognitive impairment), and the Charlson comorbidity index. A consensus was reached on the complexity criteria of the Diagnostic Instrument of Complexity in Palliative Care that could be addressed in the nursing home (no priority) and those that required a one-off (priority 2) or full (priority 1) intervention of advanced palliative care resources. Non-parametric tests were used to compare non-priority patients and patients with some kind of priority. *Results:* A high percentage of residents presented frailty (80.6%), clinical complexity (80.5%), and palliative care complexity (65.8%). A lower percentage of residents had a poor prognosis (10.1%) and an extremely poor prognosis (2%). Twelve priority 1 and 14 priority 2 elements were identified as not matching the palliative care complexity elements that had been previously identified. Of the studied cases, 20.1% had priority 1 status and 38.3% had priority 2 status. Residents with some kind of priority had greater levels of dependency (*p* < 0.001), cognitive impairment (*p* < 0.001), and poorer prognoses (*p* < 0.001). Priority 1 patients exhibited higher rates of refractory delirium (*p* = 0.003), skin ulcers (*p* = 0.041), and dyspnoea (*p* = 0.020). *Conclusions:* The results indicate that there are high levels of frailty, clinical complexity, and palliative care complexity in nursing homes. The resources available at each nursing home must be considered to determine when advanced palliative care resources are required.

## 1. Introduction

The elderly are the main population group attended to by emergency services [1]. Older individuals with multiple, complex health issues are those who most often require assistance from these services [2]. Much of the care provided to chronic patients by hospital emergency services may be considered unnecessary or delayable, or it may be provided through early intervention by primary care workers or health and social care workers [3].

Consideration should be given to whether it is appropriate to transfer an older patient to the emergency department. Emergency departments, which are usually overwhelmed, are not prepared for providing appropriate care to these patients [4]. Prolonged stays of these patients in hospital emergency departments lead to a greater number of complications and increase mortality rates among this population group [5].

It is therefore necessary to take a proactive approach in identifying elderly individuals at greater risk of experiencing adverse events (frailty) and greater synergism of conditions (complexity) in order to tailor their care accordingly and avoid unnecessary visits to emergency departments.

Nursing home residents require specific care. Increased life expectancy and higher prevalence rates of chronic conditions [6,7,8], coupled with a decline in potential informal caregivers, are causing nursing homes to become increasingly important as long-term care providers in European and Western countries [6,7].

A retrospective study in Switzerland [9] showed an annual increase in the proportion of patients from nursing homes seen in emergency departments. Similar results were obtained in a study conducted in Spain [10], which also recommended that patient characteristics should be taken into account when delivering emergency care.

Elderly individuals who are admitted to nursing homes usually require continuous, specialised care [8]. A number of studies show that these centres are home to an aged population with dependency issues, under-treated symptoms such as pain, and high prevalence rates of dementia and frailty [11,12,13,14]. In most cases, these individuals will remain in their nursing homes until they pass away, which means that they will require palliative care at some point [6,15].

It is vital to identify palliative care needs and their potential degree of complexity in these centres because this will improve the approach taken to healthcare, demands for emergency care, and use of resources [14,16]. Palliative care needs and their intensity have proven to be decisive factors in establishing priority and emergency care criteria according to patient prognosis and life expectancy [17]. 

It is essential that nursing homes and other specialised services be coordinated. Nursing homes must also reach a consensus on the criteria for referring patients with palliative care needs that cannot be adequately met in nursing homes [18,19]. The use of advanced palliative care resources (APCRs)—such as palliative care support teams or palliative care consultants—can prevent older individuals from visiting the emergency department or being admitted to hospital at the end of their lives [20].

In Andalusia, Spain, there are clear criteria in place for referring complex patients to palliative care support teams [21]. These teams, together with primary care or nursing home professionals, are jointly responsible for the care of residents with palliative care needs who meet certain criteria. These criteria, known as complexity criteria, are included in the Diagnostic Instrument of Complexity in Palliative Care (IDC-Pal) [22], which classifies cases into non-complex, complex, and highly complex, the latter two of which can be treated using APCRs.

At the same time, it is expected that non-complex residents with palliative care needs can be cared for in nursing homes without having to resort to APCRs because nursing homes are equipped with their own staff and material resources. Consequently, APCRs will be used only in complex palliative care cases. 

A number of studies have been conducted to determine the degree of frailty and complexity of residents with palliative care needs at nursing homes. However, the characteristics of the cases that are likely to require assistance from the health system’s APCRs, depending on the resources available at the nursing home, have not yet been identified.

Our hypothesis is that there are high prevalence rates of frailty, clinical complexity, and palliative care complexity among nursing home residents with chronic conditions, who may require the assistance of APCRs depending on the centre’s available resources and the degree of palliative care complexity in question. 

Therefore, the objectives of this study are to determine the frailty, prognosis, complexity, and palliative care complexity of nursing home residents with palliative care needs and define the characteristics of the cases eligible for receiving advanced palliative care according to the resources available at each nursing home.

## 2. Materials and Methods

### 2.1. Design

This was a multi-centre, descriptive, and cross-sectional study.

### 2.2. Sample

Eight nursing homes in Andalusia, Spain were selected using a convenience sampling method based on their institutional characteristics—the presence of a multi-disciplinary team, the potential involvement of professionals, and the presence of both public and private beds. All the centres included in the study had more than 60 beds. In Andalusia, nursing homes with more than 60 beds are required to offer 24-h nursing services and their own medical care [23].

In each centre, one or two nurses with intimate knowledge of patients who had been working at the nursing home for at least six months were responsible for data collection. All of the participating nurses signed an informed consent form and received training prior to data collection. 

From all the patients with dementia, chronic heart disease, chronic kidney disease, chronic liver disease, and cancer, each nursing home nurse randomly recruited 17 or 18 patients with palliative care needs based on the NECPAL CCOMS-ICO© 3.0 instrument (Instituto Catalán de Oncología, Cataluña, Spain) [8]. Patients who were reaching the end of their lives during data collection were excluded. Data were collected between June 2019 and January 2020. All participants, patients, and representatives of cognitively impaired patients were fully informed and signed an informed consent form.

### 2.3. Instruments 

Nurses collected demographic and clinical information from the patients’ clinical records using a structured questionnaire that included the following tools: 

Clinical characteristics questionnaire: The questionnaire included the Spanish versions of the Charlson comorbidity index [24], the Barthel index for assessing dependency [25], and Pfeiffer’s test for assessing cognitive impairment [26].

The Frail-VIG index: The Frail-VIG (In Spanish VIG is the abbreviation for Comprehensive Geriatric Assessment) index is a questionnaire for identifying patients with frailty, which is understood as a state of vulnerability to stressors caused by limited compensatory mechanisms [27]. This questionnaire is based on the comprehensive geriatric assessment and contains 22 items, the majority of which offer a dichotomous response option. It has been used to distinguish between different patient clinical statuses: no frailty/prefrailty (<0.20 points), baseline frailty (0.20–0.35 points), moderate frailty (0.36–0.50 points), and advanced frailty (>0.50 points). The Frail-VIG index displays predictive power regarding mortality [28].

The case complexity index (CCI): The CCI was designed to identify complex cases among non-hospitalised patients so that they can be cared for at home by nurse case managers [29]. It is based on the concept of complexity as understood by Safford, Allison, and Kiefe [30] and consists of 14 items that help to identify overall case complexity, clinical complexity, and community complexity. A complex case may be defined as a case scoring ≥100 points (total complexity) provided that the overall score for clinical management complexity items (CCI1 to CCI5) is ≥50 points.

The palliative prognostic index (PPI) is a mortality prediction instrument that has been adapted for use in nursing home residents with advanced medical conditions [31]. The PPI has been used to predict the mortality rate of patients with advanced medical conditions within the following six months (180 days). It evaluates five prognostic factors—impaired functionality in performing basic activities, delirium, severe dyspnoea, oedema, and low oral intake. This instrument allows the chances of survival to be estimated based on the score obtained. In this study, we took into consideration the positive predictive value for advanced disease within the following six months using the recalibrated version of the PPI [31]. 

The Palliative Performance Scale (PPS) [32], which is included in the PPI, assesses basic activities by assigning functionality percentages to patients ranging from 0% (deceased patient) to 100% (no sign of disease). The instrument indicates the care provision required by the patient based on the percentage assigned: no need for special care (80–100%), need for some type of care (50–70%), and need for care equivalent to hospitalisation or institutionalisation (0–40%).

The Diagnostic Instrument of Complexity in Palliative Care (IDC-Pal): The IDC-Pal is designed as a discriminating tool for complex patients who require the use of APCRs under the Andalusian Palliative Care Plan [21]. The instrument identifies 20 complexity elements and 15 high complexity elements based on the characteristics of the patient, the family, the social environment, and the healthcare organisation, and classifies cases into non-complex, complex—when there is at least one complexity element—and highly complex—when there is at least one high complexity element [22]. 

Priority levels: Using the IDC-Pal instrument and on the basis of the resources available at each nursing home, nurses at the participating nursing homes were asked to identify situations that cannot be addressed at their nursing homes and thus require immediate use of or immediate referral to APCRs (priority 1), as well as to identify situations that can be partially addressed at their nursing homes with support and intervention from APCRs (priority 2).

Once their responses had been received, the situations identified as priority 1 by at least 50% of the nurses were classified as ‘priority 1’, and the situations identified as priority 2 by at least 50% of the nurses were classified as ‘priority 2’. The rest of the situations that the nurses reported that they could deal with using the resources available at their nursing homes were classified as ‘no priority’.

### 2.4. Statistical Analysis

A descriptive analysis was carried out to summarise the main characteristics of the study sample. Categorical variables were expressed as absolute frequencies and percentages, and numerical variables were expressed as means and standard deviations (SDs).

Quantitative data were assessed for normality using the Kolmogorov–Smirnov test. All the quantitative data collected displayed a non-normal distribution (*p* < 0.001). As a result, non-parametric inferential tests were used.

Pearson’s chi-squared test was used to assess differences between groups. McNemar’s test was used to compare prevalence rates. The Mann–Whitney *U*-test was used to compare the characteristics of ‘priority 1’ and ‘priority 1–2’ groups with the ‘no priority’ group. Statistical analyses were conducted using the statistical package IBM SPSS v.24. (IBM, Armonk, NY, USA). *p*-values lower than 0.05 were considered to be statistically significant.

### 2.5. Ethical Considerations

All the participants signed an informed consent form. The study was approved by the Research Ethics Committee of the Andalusian Public Health System (reference number: AP-0105-2016 signed in 28 June 2017). The confidentiality and anonymity of the patients were preserved at all times in compliance with the Spanish Basic Law 41/2002 (Article 16). 

## 3. Results

### 3.1. Characteristics of the Sample

A total of 149 patient cases were analysed, of whom 100 (67.50%) were women. The mean age was 84.468 years (SD = 9.126). Of the groups analysed, the most frequent condition was dementia, (*n* = 67; % = 45.0), followed by chronic heart disease (*n* = 57; % = 38.3). The mean Charlson comorbidity index value was 2.587 (SD = 1.989). The mean Barthel index value was 50.369 (SD = 32.174). The average number of errors in Pfeiffer’s test was 4.9128 (SD = 3.876). 

### 3.2. Frailty

The mean Frail-VIG index was 0.2813 (SD = 0.07329). According to the Frail-VIG index results, there were 29 cases (19.46%) with no frailty, 91 cases (61.07%) with baseline frailty, and 29 cases (19.46%) with moderate frailty. The rest of the Frail-VIG index results are shown in Table 1.

### 3.3. Prognosis

As regards prognosis, the mean PPI score was 2.483 (SD = 2.571). Of the residents under study, 10.1% (*n* = 15) had a poor prognosis, which, according to the instrument’s own instructions, meant that 53% of patients with similar characteristics would die within the next six months. Three residents (2%) had an extremely poor prognosis, which meant that 68% of patients with similar characteristics would die within the next six months. Within this index, the mean PPS value was 64.429 (SD = 18.431) and the most frequent value was 50 (*n* = 44; % = 29.5). The rest of the PPI results are shown in Table 2.

### 3.4. Case Complexity

The mean CCI score was 64.1946 (SD = 14.94435) and the mean total complexity index score was 103.9933 (SD = 20.17642). Of the studied cases, 80.5% (*n* = 120) were clinically complex to manage, and 66.4% (*n* = 99) had total complexity. Table 3 shows the remaining CCI results.

### 3.5. The Diagnostic Instrument of Complexity in Palliative Care (IDC-Pal) and Priority Levels

The mean number of complexity elements identified by professionals was 0.926 (SD = 1.097), and the mean number of high complexity elements was 0.382 (SD = 0.731). Of the residents under study, 43.0% (*n* = 64) were classified as complex, and 22.8% were classified as highly complex (*n* = 34).

Based on the IDC-Pal items, 12 elements were identified as ‘priority 1’ and 14 elements were identified as ‘priority 2’. The levels of complexity defined by the IDC-Pal do not match the priority levels assigned by the nurses at the nursing homes. Of the 15 high complexity elements established by the IDC-Pal, seven were identified as ‘priority 1’, two as ‘priority 2’, and the rest were not assigned a priority (‘no priority’, NP). Of the 20 complexity elements established by the IDC-Pal, five were identified by the nurses at the nursing homes as ‘priority 1’, 12 as ‘priority 2’, and the rest were not assigned a priority (‘no priority’, NP). The remaining IDC-Pal results are shown in Table 4.

According to the priority levels established, 20.1% (*n* = 30) of the cases were to be classified as ‘priority 1’, i.e., they cannot be addressed at their nursing homes and thus require immediate use of or immediate referral to APCRs, and 38.3% (*n* = 57) of the cases were to be classified as priority 2, i.e., they can be partially addressed at their nursing homes with support and intervention from APCRs. Only 41.6% (*n* = 62) of the cases did not require any action by APCRs in the nursing home professionals’ opinion.

### 3.6. Characteristics of the Residents Classified as Priority Residents

Compared to non-priority residents, ‘priority 1’ residents exhibited higher levels of cognitive impairment (*p* = 0.007) and frailty (*p* = 0.004) and poorer prognoses as measured by the PPI (*p* = 0.000). In addition, compared to non-priority residents, residents classified as either ‘priority 1’ or ‘priority 2’ had higher levels of cognitive impairment (*p* < 0.001), poorer prognoses (*p* < 0.001), and higher dependency levels as measured by the Barthel index (*p* < 0.001). In contrast, both groups of residents showed similar levels of frailty (*p* = 0.296). The results of the comparison between the characteristics of non-priority residents and priority residents are shown in Table 5.

When comparing priority 1 residents with non-priority residents, priority 1 residents had higher rates of social vulnerability (VIG5: *p* = 0.005), refractory delirium (VIG6.1 and PPI5: *p* = 0.003), pressure and/or vascular ulcers (VIG6.3 and CCI3: *p* < 0.041), and severe dyspnoea (VIG7.2: *p* = 0.006; PPI4: *p* = 0.020).

Compared to non-priority residents, residents classified as either priority 1 or priority 2 had higher rates of cancer (*p* = 0.012), dementia (*p* < 0.001), social vulnerability (VIG5: *p* = 0.048), and refractory delirium (VIG6.1: *p* = 0.027; PPI5: *p* = 0.027), and needed more help using the telephone (VIG1.2: *p* < 0.001). 

## 4. Discussion

The results of our study suggest that there are high levels of frailty, complexity, and palliative care complexity among nursing home residents. Professionals indicated that, based on the priority levels assigned, the immediate use of APCRs was necessary in many cases. Patients eligible for receiving care from these resources had higher levels of cognitive impairment and frailty, poorer functional status, and poorer prognoses than non-priority patients. Residents with refractory delirium, severe dyspnoea, and skin ulcers were also considered a priority for receiving advanced palliative care.

In consonance with our hypothesis, the results show that approximately 80% of residents have some type of frailty. This is a considerable percentage, which nonetheless falls short of Kojima’s [33] estimate of 92.5% in one meta-analysis, perhaps as a result of the measuring instruments used. However, in the Frail-VIG validation study, Amblàs-Novellas et al. [28] reported a proportion of frail palliative patients (92.5%) that was also higher than our study, although their study was conducted among geriatric patients at an acute care unit. These percentages are particularly important because they constitute a priority criterion for using APCRs. It should be noted that the Frail-VIG index, besides being one of the few tools available in Spanish, has great predictive capacity for mortality [34], which makes it a very useful instrument for identifying patients whose needs require closer monitoring. 

Regarding complexity, 66.4% of the cases analysed were complex according to the CCI. As with frailty, the prevalence of complex cases among the general population fluctuates from 5% [35] to 24% [36], depending on how the concept of complexity is understood and the instrument used to identify it. In the instrument used in this study, complexity goes beyond co-morbidity and co-occurrence of multiple health conditions and is understood as the synergism of multiple clinical, psychological, and social problems [29]. The prevalence rate for complexity found in the nursing homes in our study was higher than the percentage of complex cases identified in the validation study for the instrument (47.3%) [29]. However, it should be borne in mind that this study was conducted with the general population and not with an institutionalised population. Not surprisingly, Amblàs-Novellas et al. [37] found that institutionalised older individuals had greater levels of co-morbidity and complexity than people of the same age living in the community. 

With regard to palliative care complexity, 43% of residents in our sample were classified as complex and 22.8% as highly complex. These complexity levels are lower than those reported by Salvador-Comino et al. [22], who used the IDC-Pal to identify complexities among patients receiving palliative care at home (32.4% of them were complex and 67.5% of them were highly complex), showing a different palliative care complexity pattern at nursing homes. However, in a study using the same instrument but with palliative patients at a haemodialysis unit [38], the percentage of complex patients (38%) was closer to the results of our study.

One of the most innovative aspects of this study is that the nursing homes’ own resources have been taken into account when identifying the priority level of residents requiring the intervention of APCRs.

The characteristics of the residents classified as priority residents do not always match the elements identified as complex in the IDC-Pal. In fact, there are many highly complex elements included in the IDC-Pal which are not particularly difficult for nursing home workers or which may be solved simply by patients staying at a nursing home—especially those derived from the patient’s family situation. Conversely, some complexity elements, such as difficulties in managing drugs and interventions, are identified by nursing home professionals as a high priority. Therefore, setting priorities for using external resources based on the response capacity of the nursing home itself can be an effective strategy for identifying intervention needs in a targeted, personalised way. This also justifies the need to adapt complexity detection instruments to this type of institution, as well as to take into account the opinions of nursing home professionals in order to decide which situations are particularly difficult and which are not.

Our results indicate that residents classified as a priority have higher levels of cognitive impairment. This is not surprising n because nursing home staff face severe difficulties in caring for individuals with dementia at the end of their lives, especially in recognising palliative care needs, pain, verbal and non-verbal communication, and behavioural disorders [39].

Furthermore, the results of our study indicate that residents with refractory symptoms such as delirium, dyspnoea, and skin ulcers are more difficult to manage in nursing homes.

With regard to delirium, according to the VIG-Frailty index, 22.1% of the residents in our study experienced this condition, which was far higher than the percentage reported in other studies [40], although this may be attributed to the different instruments used. No specific uniform tools were used to diagnose delirium in any of the nursing homes participating in this study. Special attention should be paid to improving the identification and management of delirium in nursing homes because this condition causes great stress to nursing home professionals [41]. In a retrospective study on nursing home residents’ last month of life, Smedbäck et al. [42] showed that while the pain was relieved in 43% of cases, delirium was controlled in only 4.3%. 

In this regard, the high percentage of residents in our study who regularly use psychotropic drugs (81.2%) is striking. This percentage is higher than those reported by other studies in the same context [43]. This may be explained both by the high percentage of patients with dementia and delirium in our study and by the fact that no distinction was made during data collection between the different types of drugs (antidepressants, antipsychotics, anxiolytics, etc.), as in other studies [43]. These data must be taken into account because the use of psychotropic drugs may be a factor contributing to the development of adverse effects, hospitalisation, and even death [44].

For nursing home professionals, dyspnoea is another symptom that is most difficult to control. In the aforementioned study [42], dyspnoea was only controlled in 6.1% of cases. Due to the high prevalence of advanced dementia, dyspnoea caused by aspiration pneumonia is a common symptom causing great stress for professionals and suffering for residents [45]. On many occasions, dyspnoea crises require specific medication and equipment which are sometimes only available in hospitals. Therefore, staff training and material resources are required in nursing homes.

There was a 12.1% prevalence rate of skin ulcers in our study, falling within previously published prevalence rates, which ranged from 3.4% to 32.4%, depending on the criteria used to identify them [46]. In our study, priority 1 residents had a higher proportion of skin ulcers than non-priority residents. It should be noted that pressure ulcers are very costly for nursing homes in terms of equipment and time spent caring for them [47], so to ensure adequate treatment, professionals may need the support of other resources within the health system. 

As measured by the PPI results, higher priority levels indicate poorer prognoses among residents. According to a recent study conducted in our setting, residents admitted to nursing homes have a higher prevalence of symptoms in their last month of life and are also subject to a greater number of interventions that are not necessarily palliative [48]. It is thus essential to have the support of APCRs when a patient’s prognosis is very poor because they prevent patients from being transferred to a hospital to control their symptoms in the final moments of their lives [20]. 

Our study has a number of limitations that should be taken into account. To ensure their participation in the study, both nursing homes and professionals were selected using a convenience sampling method, which could have introduced selection bias. In addition, this study was carried out with residents with palliative care needs, as identified using the NECPAL-ICO-OMS instrument (Instituto Catalán de Oncología, Cataluña, Spain), whose characteristics do not necessarily reflect those of the entire population institutionalised in nursing homes. Finally, this study used instruments validated directly in Spanish, but it is important to emphasise that the proportion of residents experiencing frailty, complexity, or palliative care complexity may vary depending on the tool used to identify them.

## 5. Conclusions

In conclusion, there are high levels of frailty, clinical complexity, and palliative care complexity among nursing home residents. There is a need for nursing home professionals to identify which situations constitute a priority for the use of APCRs because these situations do not necessarily coincide with the tools for assessing complexity used in other settings. Our results indicate that symptom control, emergency situations in cancer patients, and managing complex drugs and interventions receive higher levels of priority. Residents classified as a priority according to these criteria display higher levels of cognitive impairment and frailty, poorer prognoses and functional status, and higher rates of refractory delirium, severe dyspnoea, and skin ulcers.

## Figures and Tables

**Table 1 medicina-57-00070-t001:** Results of the Frail-VIG index (*n* (%)).

	NO		YES	
VIG1.1 Needs help managing financial matters (bank, shops, restaurants)	22 (14.8%)		127 (85.2%)	
VIG1.2 Needs help using the telephone	64 (43.0%)		85 (57.0%)	
VIG1.3 Needs assistance in preparing or administering medications	11 (7.4%)		138 (92.6%)	
VIG1.4 Barthel index (dependency)	No dependency (BI ≥ 95)	Mild-moderate dependency (BI 90–65)	Moderate-severe dependency (BI 60–25)	Absolute dependency (BI ≤ 20)
	17 (11.4%)	46 (30.9%)	50 (33.6%)	36 (24.2%)
VIG3. Degree of cognitive impairment	No cognitive impairment	Mild-moderate (GDS ≤ 5 *)	Severe-very severe (GDS ≥ 6 *)	
	65 (43.6%)	37 (24.8%)	47(31.5%)	
	NO		YES	
VIG2. Weight loss ≥5% in the last 6 months	136 (91.3%)		13 (8.7%)	
VIG4.1 Need for antidepressant medication	28 (18.8%)		121 (81.2%)	
VIG4.2 Frequent need for benzodiazepines or other psychiatric drugs with a sedative effect for insomnia/anxiety	28 (18.8%)		121 (81.2%)	
VIG5 Do healthcare professionals perceive the presence of social vulnerability?	123 (82.9%)		26 (17.4%)	
VIG6.1 Presence of delirium and/or behaviour disorder requiring antipsychotic drugs in the last six months	116 (77.9%)		33 (22.1%)	
VIG6.2 In the last six months, ≥2 falls or hospitalisation due to a fall	124 (83.2%)		25 (16.8%)	
VIG6.3 Presence of ulcer (pressure or vascular, any grade)	131 (87.9%)		18 (12.1%)	
VIG6.4 Taking ≥5 drugs	25 (16.8%)		124 (83.2%)	
VIG6.5 Difficulty swallowing when eating or drinking? Presence of aspiration respiratory infections during the last six months?	138 (92.6%)		11 (7.4%)	
VIG7.1 Need for ≥2 conventional analgesics and/or strong opioids for pain control	113 (75.8%)		36 (24.2%)	
VIG7.2 Basal dyspnoea impeding the ability to leave the house and/or opioids are frequently needed	142 (95.3%)		7 (4.7%)	
VIG8.1 Active cancer	125 (83.9%)		24 (16.1%)	
VIG8.2 Presence of any type of chronic respiratory disease (COPD, restrictive lung disease, etc.)	115 (77.2%)		34 (22.8%)	
VIG8.3 Presence of any type of chronic heart disease (heart failure, ischemic cardiomyopathy, arrhythmia)	92 (61.7%)		57 (38.3%)	
VIG8.4 Presence of any type of neurodegenerative disease (Parkinson, ALS, etc.) or a history of stroke (ischemic or haemorrhagic)	120 (80.5%)		29 (19.5%)	
VIG8.5 Presence of any type of chronic digestive disease (chronic liver disease, cirrhosis, chronic pancreatitis, inflammatory bowel disease, etc.)	148 (99.3%)		1 (0.7%)	
VIG8.6 Presence of chronic renal failure	130 (87.2%)		19 (12.8%)	

* GDS = Reisberg’s Global Deterioration Scale (Auer, S.; Reisberg, B. The GDS/FAST staging system. *Int. Psychogeriatry*
**1997**, *9*, 167–171). VIG: In Spanish VIG is the abbreviation for Comprehensive Geriatric Assessment. BI: Barthel index. COPD: Chronic Obstructive Pulmonary Disease. ALS: Amyotrophic Lateral Sclerosis.

**Table 2 medicina-57-00070-t002:** Results of the palliative prognosis index (PPI) (*n* (%)).

	≥60	30–50 Points				10–20 Points
**PPI1—PPS**	94 (63.1%)	54 (36.2%)				1 (0.7%)
**PPI2—Oral intake**	Normal	Moderately reduced				Severely reduced
	114 (76.5%)	33 (22.1%)				2 (1.3%)
	No			Yes		
**PPI3—Oedema**	116 (77.9%)			33 (22.1%)		
**PPI4—Dyspnoea at rest**	141 (94.6%)			8 (5.4%)		
**PPI5—Delirium**	116 (77.9%)			33 (22.1%)		
	>0 40% of patients with similar characteristics would die within the next six months	>2 42% of patients with similar characteristics would die within the next six months	>4 47% of patients with similar characteristics would die within the next six months		>6 53% of patients with similar characteristics would die within the next six months	>9.5 68% of patients with similar characteristics would die within the six months
**Overall PPI score**	72 (48.3%)	50 (33.6%)	9 (6.0%)		15 (10.1%)	3 (2%)

PPS: Palliative Performance Status.

**Table 3 medicina-57-00070-t003:** Results of the case complexity index (CCI) (*n* (%)).

CCI 1—Severity: Severity Level	Level 2: Has Severe Mental Disorders or One Severe Advanced Organic Disease (Grades III–IV on Any Scale)		Level 3: Needs Palliative Care (Based on an ICD Identification Code or a Specific Scale or Report)	
	6 (4.0%)		143 (96.0%)	
	NO		YES	
CCI2—Polypathology: ≥2 organic systems (cardiovascular, renal, respiratory, digestive, nervous, endocrine, haematological, osteoarticular, etc.) affected by chronic disease	42 (28.2%)		107 (71.8%)	
CCI3—Skin: Skin ulcers	131 (87.9%)		18 (12.1%)	
	No admissions	1 hospital admission through the emergency department in the last year		≥2 hospital admissions or 1 admission to a home support team or to a hospital for the chronically ill in the last year
CCI4—Admissions	108 (72.5%)	24 (16.1%)		17 (11.4%)
	NO		YES	
CCI5—≥2 visits to a hospital emergency department without the patient being admitted in the last 12 months	109 (73.2%)		40 (26.8%)	
CCI6—Polypharmacy: ≥5 chronic drugs (during ≥6 months) or ≤4 chronic drugs ineffectively managed by both patient and caregiver	25 (16.8%)		124 (83.2%)	
CCI7—Technology: Requires support for ≥1 vital functions at home: breathing, nutrition, and/or elimination (e.g., supplemental oxygen, mechanical ventilation, enteral or parenteral nutrition, ostomies, dialysis, bladder catheterisation)	124 (83.2%)		25 (16.8%)	
CCI8—Technical supports: Requires or has an anti-bedsore mattress, an adjustable bed, a crane, or a wheelchair	62 (41.6%)		87 (58.4%)	
CCI9—Dependency	Level 1: Moderate or severe dependency regarding basic activities of daily living (Barthel 20/55) or cognitive impairment (Pfeiffer 4–7 points)		Level 2: Absolute dependency regarding basic activities of daily living (Barthel ≤ 15) or dementia or cognitive impairment (Pfeiffer ≥ 8 points)	
	27 (18.1%)		122 (81.9%)	
	NO		YES	
CCI10 ≥ 2 unexplained falls, an unexplained fall resulting in a fracture, or hospital admission in the last six months	124 (83.2%)		25 (16.8%)	
CCI11 Lives alone without caregivers or caregivers have limited ability to support the patient at home or experience difficulties in doing so.	149 (100%)		0 (0%)	
CCI12 Architectural barriers (e.g., in doorways, upper floors without lifts, inside the home, etc.), poor housing, or geographical isolation.	141 (94.6%)		8 (5.4%)	
CCI13 Age (≥75 or ≤15 years old), has no education, or does not speak the language or gis/her culture (ethnicity, religion, etc.) which hinders/prevents the intervention.	55 (36.9%)		94 (63.1%)	
CCI14 Family conflicts, lack of financial resources, or suspected abuse	129 (86.6%)		20 (13.4%)	
CLINICAL COMPLEXITY (CCI1–5 ≥ 100)	29 (19.5%)		120 (80.5%)	
TOTAL COMPLEXITY (score ≥ 150)	50 (33.6%)		99 (66.4%)	

**Table 4 medicina-57-00070-t004:** Results of the Diagnostic Instrument of Complexity in Palliative Care (IDC-Pal) complexity elements, IDC-Pal high complexity elements, and priority levels (*n* (%)).

		NO	YES	C/HC	Priority Level	Relationship between IDC-Pal Complexity and Priority Level	
PATIENT: HISTORY	IDCPAL1.1a The patient is a child or adolescent.	149 (100%)	0 (0%)	HC	P1	Match	C = P
	IDCPAL1.1b The patient is a healthcare professional.	149 (100%)	0 (0%)	C	NP	No match	C > P
	IDCPAL1.1c Social-family role performed by the patient	144 (96.6%)	5 (3.4%)	C	NP	No match	C > P
	IDCPAL1.1d Previous physical, psychological, or sensorial disability	135 (90.6%)	14 (9.4%)	C	P2	Match	C = P
	IDCPAL1.1e Recent and/or active addiction problems	145 (97.3%)	4 (2.7%)	C	P2	Match	C = P
	IDCPAL1.1f Previous mental illness	144 (96.6%)	5 (3.4%)	C	P2	Match	C = P
PATIENT: CLINICAL SITUATION	IDCPAL1.2a Symptoms difficult to control	139 (93.3%)	10 (6.7%)	HC	P1	Match	C = P
	IDCPAL1.2b Refractory symptoms	147 (98.7%)	2 (1.3%)	HC	P1	Match	C = P
	IDCPAL1.2c Urgent situations in the terminal cancer patient	149 (100%)	0 (0%)	HC	P1	Match	C = P
	IDCPAL1.2d Last hours/days of life difficult to control	148 (99.3%)	1 (.7%)	HC	P1	Match	C = P
	IDCPAL1.2e Clinical situations due to cancer progression difficult to control	148 (99.3%)	1 (.7%)	HC	P1	Match	C = P
	IDCPAL1.2f Acute decompensated organ insufficiency in non-oncological terminal patient	110 (73.8%)	39 (26.2%)	C	P2	Match	C = P
	IDCPAL1.2g Severe cognitive failure	141 (94.6%)	8 (5.4%)	C	P1	No Match	C < P
	IDCPAL1.2h Abrupt change in level of functional autonomy	143 (96.0%)	6 (4.0%)	C	NP	No Match	C > P
	IDCPAL1.2i Presence of comorbidity difficult to control	149 (100%)	0 (0%)	C	P2	Match	C = P
	IDCPAL1.2j Severe constitutional syndrome	146 (98.0%)	3 (2.0%)	C	P2	Match	C = P
	IDCPAL1.2k Clinical management difficult due to repeated non-compliance with therapy	149 (100%)	0 (0%)	C	P2	Match	C = P
PATIENT: PSYCHO-EMOTIONAL SITUATION	IDCPAL1.3a Risk of patient committing suicide	149 (100%)	0 (0%)	HC	P2	No Match	C > P
	IDCPAL1.3b Patient is asking to hasten the process of death	142 (95.3%)	7 (4.7%)	HC	P2	No Match	C > P
	IDCPAL1.3c Patient presents existential anguish and/or spiritual suffering	144 (96.6%)	5 (3.4%)	HC	NP	No Match	C > P
	IDCPAL1.3d Communication conflicts between patient and family	146 (98.0%)	3 (2.0%)	C	P2	Match	C = P
	IDCPAL1.3e Communication conflicts between patient and healthcare team	144 (96.6%)	5 (3.4%)	C	P2	Match	C = P
	IDCPAL1.3f Inadequate emotional coping by patient	126 (84.6%)	23 (15.4%)	C	P2	Match	C = P
FAMILY AND ENVIRONMENT	IDCPAL2.a Absent or insufficient family support and/or caregivers	142 (95.3%)	7 (4.7%)	HC	NP	No Match	C > P
	IDCPAL2.b Family members and/or caregivers not competent to give care	147 (98.7%)	2 (1.3%)	HC	NP	No Match	C > P
	IDCPAL2.c Dysfunctional family	134 (89.9%)	15 (10.1%)	HC	NP	No Match	C > P
	IDCPAL2.d Family and/or caregiver burden	147 (98.7%)	2 (1.3%)	HC	NP	No Match	C > P
	IDCPAL2.e Complex bereavement	143 (96.0%)	6 (4.0%)	C	P1	No Match	C < P
	IDCPAL2.f Structural limitations of environment for the patient	148 (99.3%)	1 (0.7%)	HC	NP	No Match	C > P
PROFESSIONALS/TEAM	IDCPAL3.1. Application of palliative sedation difficult to manage	144 (96.6%)	5 (3.4%)	HC	P1	Match	C = P
	IDCPAL3.2. Difficulty in the indication and/or management of medication	145 (97.3%)	4 (2.7%)	C	P1	No Match	C < P
	IDCPAL3.3. Difficulty in the indication and/or management of interventions	147 (98.7%)	2 (1.3%)	C	P1	No Match	C = P
	IDCPAL3.4. Limitations of professional competence to address situations	147 (98.7%)	2 (1.3%)	C	P2	Match	C = P
RESOURCES	IDCPAL3.5. Difficulty managing or acquiring instrumental techniques and/or specific material at home	142 (95.3%)	7 (4.7%)	C	P1	No Match	C < P
	IDCPAL3.6. Difficulty managing coordination and logistic needs	149 (100%)	0 (0%)	C	P2	Match	C = P

C = complexity; HC = highly complex; P1 = priority 1; P2 = priority 2; NP = no priority; C = P = complexity level matches priority level; C > P = complexity level higher than priority level; and P > C = priority level higher than complexity level.

**Table 5 medicina-57-00070-t005:** Comparison between the characteristics of non-priority and priority residents (Mann–Whitney *U*-test).

	NP *n* = 62		P1 + P2 *n* = 87		NP vs. P1 + P2	P1 *n* = 30		NP vs. P1
	M	SD	M	SD	*p*	M	SD	*p*
Age	85.468	8.751	83.759	9.370	0.111	82.500	10.051	0.131
Barthel	62.541	26.874	43.412	32.571	0.001 *	48.966	33.123	0.065
Pfeiffer	2.950	2.896	6.379	3.831	0.001 *	5.066	3.885	0.007
Charlson	2.459	1.831	2.678	2.099	0.863	2.933	2.348	0.566
VIG	0.274	0.083	0.286	0.066	0.296	0.309	0.063	0.028
PPI	1.516	1.890	3.172	2.773	0.001 *	3.900	2.884	0.001 *
CCI Clin	65.403	14.551	63.333	15.244	0.498	63.500	17.673	0.296
Overall CCI score	105.242	20.374	103.103	20.105	0.411	105.167	23.211	0.923

*: *p* < 0.001.

## Data Availability

The data presented in this study are available on reasonable request from the corresponding author. The data are not publicly available due to privacy restrictions.

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
