# Peer review of "Frailty, Complexity, and Priorities in the Use of Advanced Palliative Care Resources in Nursing Homes"

_medicina, 2021, doi:10.3390/medicina57010070_

Round 1

Reviewer 1 Report

This is a descriptive cross-sectional study of 149 residents of Spanish nursing homes. Using several established methods validated in Spain the aim was to identify the characteristics that indicate the need for APCRs (advanced palliative care resources). In particular, the authors refer in this context to the importance to avoid non-indicated emergency room visits and hospital admissions, respectively, for nursing home residents in need of palliative care.

General Comment:

The image obtained of the study population based on the individual items in the results section appears realistic for nursing home residents with a mean age of 84.5 years. The conclusion that the studied patients are highly affected by frailty and palliative complexity is plausible, as well as the resulting demand for the expansion of APCRs according to the requirements and the available resources directly on-site.

However, it needs a fundamental revision in my opinion. Notably, in the results section, one is almost overwhelmed by the abundance of detailed data; with the current presentation, it is difficult to keep track („I can no longer see the wood for the trees“). The authors used instruments developed in Spain that may not be consistently familiar to an international readership (thus, the frailty scale deviates significantly from the internationally used methods). This does not preclude publication of the paper per se. For readers without experience with the tools used here several of the scores given (e.g., PPI or PPS score) can hardly be interpreted. Consequently, the authors should present the results in a way that is easily understood by non-Spanish readers. Overall, I recommend checking whether the introduction could be written more concise. The results section should be more clearly structured, especially with regard to an international readership.

Special Comments:

Line 66: "overaged" - please have native speaker check: (ageist terminology?).

Line 167/168: given the large number of items recorded, should a Bonferroni correction be carried out?

Table 1, VIG3: here the abbreviation "GDS" is used without prior explanation. Is the Reisberg scale meant?

Table 1, VIG4.1/2: the high percentage of use of psychotropic drugs, especially sedatives, is striking: this should be discussed because of possible effects on other clinical variables (please note the delirogenic effect of benzodiazepines).

Table 1, VIG 8.3: "arMGRythmia" - misspelling?

Table 2: Not explained what the PPS includes.

Table 2: overall PPI score: the different time periods for which survival rates are given are confusing.

Table 3: could it be that "Yes" and "No" were mixed-up from CCI10 onwards, e. g. is it true that 83.2% of patients had experienced falls in the last 6 months?

Table 3: Architectural barriers present in 94.6%? What is the significance of this item for nursing home residents?

Table 3: Difference "clinical complexity" vs. "total complexity"?

Line 248: "peMGHaps" or "perhaps"?

Line 294/295: the authors discuss the observed delirium prevalence of 22.1%. This value does not seem inconceivable given the high degree of comorbidity and impairment in the study population. Question: which method was used to diagnose the delirium and which time period was used as a basis.

Line 318 following: the authors correctly discuss the limitations of their study. I propose a reminder on the recruitment of the sample ("convenience sample", as already mentioned in paragraph 2.2).

Author Response

Reviewer 1

Reviewer (R): This is a descriptive cross-sectional study of 149 residents of Spanish nursing homes. Using several established methods validated in Spain the aim was to identify the characteristics that indicate the need for APCRs (advanced palliative care resources). In particular, the authors refer in this context to the importance to avoid non-indicated emergency room visits and hospital admissions, respectively, for nursing home residents in need of palliative care.

General Comment:

The image obtained of the study population based on the individual items in the results section appears realistic for nursing home residents with a mean age of 84.5 years. The conclusion that the studied patients are highly affected by frailty and palliative complexity is plausible, as well as the resulting demand for the expansion of APCRs according to the requirements and the available resources directly on-site.

However, it needs a fundamental revision in my opinion. Notably, in the results section, one is almost overwhelmed by the abundance of detailed data; with the current presentation, it is difficult to keep track („I can no longer see the wood for the trees“). The authors used instruments developed in Spain that may not be consistently familiar to an international readership (thus, the frailty scale deviates significantly from the internationally used methods). This does not preclude publication of the paper per se. For readers without experience with the tools used here several of the scores given (e.g., PPI or PPS score) can hardly be interpreted. Consequently, the authors should present the results in a way that is easily understood by non-Spanish readers. Overall, I recommend checking whether the introduction could be written more concise. The results section should be more clearly structured, especially with regard to an international readership.

Authors (A): Thanks for your kind commentaries and suggestions. We have re-restructured the results and methods section to improve its reading. In methods section, several information regarding tools has been included to facilitate understanding for non-Spanish readers.

Regarding VIG-Frailty Scale:

Line 129: It has been used to discriminate between different patient clinical statuses: Absence of frailty/prefrailty (<0.20 points), baseline frailty (0.20-0.35 points), Intermediate frailty (0.36-0.50 points) y advanced frailty (>0.50 points). VIG Frailty Scale has shown a predictive power regarding mortality [28].

Regarding CCI

Line 136: A complex case is defined as one that has a score of ≥100 points, provided that the score between the items of Clinical Management Complexity (CCI1 to CCI5) is ≥50 points.

Regarding PPI and PPS

Line 142: Five prognostic factors are evaluated: deterioration of functionality for the development of basic activities, delirium, severe dyspnoea, and low oral intake. This instrument allows to estimate an approximate survival based on the score obtained. For this study, the positive predictive value of the recalibrated version for advanced diseases at six months has been considered.

PPS [32], which is included in the PPI, assesses basic activities assigning a percentage of functionality ranging from 0% (deceased patient) to 100% (normal, no signs of disease). Depending on the percentage in which the patient is assigned, the instrument indicates the provision of care: no need for special care (80-100%), need for some type of care (50-70%) and care equivalent to hospitalisation or institutionalisation (0-40%).

Special Comments:

(R): Line 66: "overaged" - please have native speaker check: (ageist terminology?).

(A): Thanks for your suggestion. The term overaged has been replaced by aged.

Line 66 Various studies show that these centres are home to an aged population with dependency issues, under-treated symptoms such as pain, and high prevalence rates of dementia and frailty [11-14]

(R): Line 167/168: given the large number of items recorded, should a Bonferroni correction be carried out?

(A): The Bonferroni correction adjusts probability (p) values because of the increased risk of a type I error when making multiple statistical tests. Bonferroni is usually used in analysis of variance in parametric (ANOVA-Test) and non-parametric (Kruskal-Wallis) tests. In our study we used McNemar’s test to compare prevalence rates and Mann-Whitney U-test to compare the characteristics of “Priority 1” and “Priority 1-2” groups with the “No priority” group. The use of Bonferroni correction in this kind of test is controversial (See for example: Armstrong RA. When to use the Bonferroni correction. Ophthalmic Physiol Opt. 2014 Sep;34(5):502-8. doi: 10.1111/opo.12131.) so we are reluctant to apply this correction to our analysis. If the reviewer consider it is mandatory to apply this correction, we will apply it to our analysis.

(R): Table 1, VIG3: here the abbreviation "GDS" is used without prior explanation. Is the Reisberg scale meant?

(A): Thanks. A footnote has been added to Table 1 to clarify that GDS is Reisberg Global Deterioration Scale.

(R): Table 1, VIG4.1/2: the high percentage of use of psychotropic drugs, especially sedatives, is striking: this should be discussed because of possible effects on other clinical variables (please note the delirogenic effect of benzodiazepines).

(A): Indeed, it is a striking percentage. A new paragraph has been added in order to discuss the potential effect on other variables.

Line 315. In this respect, the high percentage of residents in our sample who consume psychotropic drugs (81.2%) is striking. This percentage is higher than that of other studies in our context [43]. This may be due not only to the high percentage of patients with dementia and delirium in our study, as well as the fact that in the collection of data no distinction has been made between the different types of drugs (antidepressants, antipsychotics, anxiolytics, etc), as has been done in other studies [43]. In any case, it is necessary to take these data into account, as the use of psychotropic drugs can be a factor that contributes to the development of adverse effects, hospitalisation and even death [44].

Two new references have been added to the reference section.

  1. Janus, S.I.; van Manen, J.G.; IJzerman, M.J.; Zuidema, S.U. Psychotropic drug prescriptions in Western European nursing homes. Int Psychogeriatr. 2016, 28(11), 1775-1790.
  2. Lapeyre-Mestre, M. A Review of Adverse Outcomes Associated with Psychoactive Drug Use in Nursing Home Residents with Dementia. Drugs Aging. 2016, 33(12), 865-888.

(R): Table 1, VIG 8.3: "arMGRythmia" - misspelling?

(A): Thanks. This is a misspelling. The correct term is arrhythmia. It has been changed in Table 1

(R): Table 2: Not explained what the PPS includes.

(A): A short explanation has been included in methods section:

Line 147: PPS [32] included in the IPP which assesses basic activities, assigns a percentage of functionality ranging from 0% (deceased patient) to 100% (normal, no signs of disease). Depending on the percentage in which the patient is, the instrument provides an indication of the provision of care, differentiating between: no need for special care (80-100%), need for some type of care (50-70%) and care equivalent to hospitalisation or institutionalisation (0-40%).

(R): Table 2: overall PPI score: the different time periods for which survival rates are given are confusing.

(A): This tool reports predictive value according to the percentage of patients with similar characteristics that would die in a time period. The overall PPI score time periods of Table 2 have been changed to fit positive predictive values of recalibrated tool for advanced chronic disease in Spanish Version (Nieto Martín, M.D.; Bernabeu Wittel, M.; De La Higuera Vila, L.; Mora Rufete, A.; Barón Franco, B.; Ollero Baturone, M. Recalibración del Palliative Prognostic Index en pacientes con enfermedades médicas avanzadas. Rev Clin Esp 2013, 213(7), 323–9).

Additional information has been added to methods section:

Line 142: Five prognostic factors are evaluated: deterioration of functionality for the development of basic activities, delirium, severe dyspnoea and low oral intake. This instrument allows to estimate an approximate survival based on the score obtained. For this study, the positive predictive value of the recalibrated version for advanced diseases at six months has been considered [31].

(R): Table 3: could it be that "Yes" and "No" were mixed-up from CCI10 onwards, e. g. is it true that 83.2% of patients had experienced falls in the last 6 months?

(A): Thanks. “Yes” and “No” were mixed-up in this item. It has been changed in Table 3.

(R): Table 3: Architectural barriers present in 94.6%? What is the significance of this item for nursing home residents?

(A): Thanks. “Yes” and “No” were mixed-up in this item too. It has been changed in Table 3. There are only 8 cases (5.4%) with architectural barriers.

(R): Table 3: Difference "clinical complexity" vs. "total complexity"?

(A): An additional explanation has been added to methods section to difference clinical complexity and total complexity.

Line 136: A complex case is defined as one that has a score of ≥100puntos, provided that the score between the items of Clinical Management Complexity (CCI1 to CCI5) is ≥50 points.

(R): Line 248: "peMGHaps" or "perhaps"?

(A): Thanks. This is a misspelling. The correct term is perhaps. It has been changed in Line 262

(R): Line 294/295: the authors discuss the observed delirium prevalence of 22.1%. This value does not seem inconceivable given the high degree of comorbidity and impairment in the study population. Question: which method was used to diagnose the delirium and which time period was used as a basis.

(A): The prevalence of delirium was obtained through the VIG-Frailty Scale (VIG6.1 Presence of delirium and/or behaviour disorder requiring antipsychotic drugs in the last 6 months). No consensus specific tools have been used to diagnose delirium for all the nursing homes that participated in this study. A sentence has been added to discussion section.

Line 307: With regard to delirium, according to VIG-Frailty Scale, 22.1% of the residents in our study experienced this condition, which was far higher than the percentage reported in other studies [40], although this may be attributed to the different instruments used. No consensus specific tools have been used to diagnose delirium for all the nursing homes that participated in this study.

(R): Line 318 following: the authors correctly discuss the limitations of their study. I propose a reminder on the recruitment of the sample ("convenience sample", as already mentioned in paragraph 2.2).

(A): Thanks. This information has been added to discussion:

Line 342: Our study has a number of limitations that should be taken into account. To ensure their participation in the study, both nursing homes and professionals were selected using a convenience sampling method, which could have introduced selection bias.

Reviewer 2 Report

This is an interesting study

The results are presented well but it would be very helpful to have more details of the instruments used - perhaps in the Methods- so the scores make more sense, eg have the levels of scores / minimum and maximum level

The use of "pass away" should be considered.  In a palliative care study "die" may be more appropriate

Author Response

Reviewer 2

Reviewer (R): This is an interesting study

The results are presented well but it would be very helpful to have more details of the instruments used - perhaps in the Methods- so the scores make more sense, eg have the levels of scores / minimum and maximum level

Authors (A): Thanks for your kind commentaries and suggestions. In methods section, several information regarding tools has been included to facilitate understanding

Regarding VIG-Frailty Scale:

Line 129: It has been used to discriminate between different patient clinical statuses: Absence of frailty/prefrailty (<0.20 puntos), baseline frailty (0.20-0.35 puntos), Intermediate frailty (0.36-0.50) y advanced frailty (>0.50 puntos). VIG Frailty Scale has shown a predictive power regarding mortality [28].

Regarding CCI

Line 136: A complex case is defined as one that has a score of ≥100puntos, provided that the score between the items of Clinical Management Complexity (CCI1 to CCI5) is ≥50 points.

Regarding PPI and PPS

Line 142: Five prognostic factors are evaluated: deterioration of functionality for the development of basic activities, delirium, severe dyspnoea, and low oral intake. This instrument allows us to estimate an approximate survival based on the score obtained. For this study, the positive predictive value of the recalibrated version for advanced diseases at six months has been considered.

PPS [32], which is included in the PPI, assesses basic activities, assigning a percentage of functionality ranging from 0% (deceased patient) to 100% (normal, no signs of disease). Depending on the percentage in which the patient is assigned, the instrument indicates the provision of care: no need for special care (80-100%), need for some type of care (50-70%) and care equivalent to hospitalisation or institutionalisation (0-40%).

(R): The use of "pass away" should be considered.  In a palliative care study "die" may be more appropriate

(A): The expression pass away has been replaced by “die” in table 2

Round 2

Reviewer 1 Report

General points:

The changes in the revision compared to the first version have improved the comprehensibility of the article. Nevertheless, the section “Results” is still cumbersome to read. Many aspects remain incompletely understandable to the reader, e.g. the procedure of calculation of the different scores (Frail-VIG, CCI etc.). If possible, these data should be presented in an electronic supplement. In my opinion, this would considerably increase the value of the article, in particular for non-Spanish readers being not familiar with the instruments used in this study. If this has not already been the case, the contribution should be additionally assessed by a reviewer familiar with the instruments preferably used in Spain.

Details:

Line 142/143: “Five prognostic factors are evaluated: deterioration of functionality for the development of basic activities, delirium, severe dyspnoea, and low oral intake”. Please add: “edema”.

Line 194/195: “According to the Frail-VIG, there were 29 cases (19.5%) with intermediate frailty, and 91 cases (61.1%) with baseline frailty.” There remain 29 cases without fraiIty, I assume?

Line 198: please add reference for Reisberg-scale.

Paragraph 3.3, including Table 2: The presentation is confusing and partly contradictory. In the text, survival rates are given for 3 and 4.5 months, but in the table they refer to 6 months. It is also noticeable that the survival rates for a score > 0 hardly differ from those for patients with a score > 2.

Paragraph 3.4, including Table3: here the “Total Complexity Index score” is mentioned for the first time; I do not find comments on this tool in the “Methods” section. According to item CCI1 96,0 % of the patients were in need of palliative care. This seems not congruent to other figures in the “Results” section (see line 231) and should be addressed in the discussion. CCI9: Taking into account the data in section 3.1, I would have expected that there should also have been some (few) patients without more severe impairment in the Barthel- and Pfeiffer-Score, respectively. CCI14: “Yes” in 86,6 % of the cases seems rather high.

Line 218/219: “The levels of complexity defined by the IDC-Pal do not match the priority levels assigned by the nurses at the nursing homes.” These data are not presented in detail.

Author Response

Dear editor

I would first like to thank the reviewers for their comments and suggestions. Without a doubt, they have helped improve the manuscript’s quality, have clarified the way ideas are introduced, and entail greater understanding on the part of the reader.

Reviewer 1

Reviewer (R): The changes in the revision compared to the first version have improved the comprehensibility of the article. Nevertheless, the section “Results” is still cumbersome to read. Many aspects remain incompletely understandable to the reader, e.g. the procedure of calculation of the different scores (Frail-VIG, CCI etc.). If possible, these data should be presented in an electronic supplement. In my opinion, this would considerably increase the value of the article, in particular for non-Spanish readers being not familiar with the instruments used in this study. If this has not already been the case, the contribution should be additionally assessed by a reviewer familiar with the instruments preferably used in Spain.

Authors (A): Thank you for your comments and suggestions.

For the score calculation of the different tools, we have followed the instructions of the authors.

The score calculation of the Frailty-VIG is displayed in “Amblàs-Novellas, J., Martori, J.C., Espaulella, J. et al. Frail-VIG index: a concise frailty evaluation tool for rapid geriatric assessment. BMC Geriatr 18, 29 (2018). https://doi.org/10.1186/s12877-018-0718-2”

The score calculation of the CCI could be consulted in the doctoral thesis “Evidencias de validez de un índice de complejidad de casos. Autores: María Luisa Ruiz Miralles. Directores de la Tesis: Miguel Richart Martínez (dir. tes.). 2016. Universidad de Alicante”. The developers of this tool have prepared a manuscript in English that has been accepted for publication in a peer-reviewed journal. We have added this new reference to facilitate its consultation by international readers.

Ruiz-Miralles, M.L.; Richart-Martínez, M.; García-Sanjuan, S.; Gallud Romero, J.; Cabañero-Martínez, M.J. Design and validation of a complex case evaluation index. An. Sist. Sanit. Navar. 2020 (in press).

Anyway, CCI developers are aware of and have assessed us in the preparation of this manuscript.

Details:

(R): Line 142/143: “Five prognostic factors are evaluated: deterioration of functionality for the development of basic activities, delirium, severe dyspnoea, and low oral intake”. Please add: “edema”.

(A): Thanks. Oedema has been added to the prognostic factors. Line 142

It evaluates five prognostic factors: impaired functionality in performing basic activities, delirium, severe dyspnoea, oedema, and low oral intake.

(R): Line 194/195: “According to the Frail-VIG, there were 29 cases (19.5%) with intermediate frailty, and 91 cases (61.1%) with baseline frailty.” There remain 29 cases without fraiIty, I assume?

(A): Yes. It has been added to the text. Line 194

According to the Frail-VIG, there were 29 cases with no frailty, 91 cases (61.1%) with baseline frailty, and 29 cases (19.5%) with moderate frailty.

(R): Line 198: please add reference for Reisberg-scale.

(A): A reference has been added to support the use of the Reisberg-Scale

*GDS = Reisberg’s Global Deterioration Scale (Auer, S.;, Reisberg, B. The GDS/FAST staging system. Int Psychogeriatry, 1997,9, 167-71)

(R): Paragraph 3.3, including Table 2: The presentation is confusing and partly contradictory. In the text, survival rates are given for 3 and 4.5 months, but in the table they refer to 6 months. It is also noticeable that the survival rates for a score > 0 hardly differ from those for patients with a score > 2.

(A): Thanks. The line in the text has been changed to fit the results showed in Table 2. In this study, we took into consideration the positive predictive value for advanced disease within the following 6 months using the recalibrated version of the PPI. Line 200

10.1% of the residents (n=15) had a poor prognosis, which, according to the instrument’s own instructions, meant that 53% of patients with similar characteristics would die within the next 6 months. Three residents (2%) had an extremely poor prognosis, which meant that 68% of patients with similar characteristics would die within the next 6 months.

(A): That has also been changed in the abstract. (Line 31)

10.1% had a poor prognosis and 2% an extremely poor prognosis.

(R): Paragraph 3.4, including Table3: here the “Total Complexity Index score” is mentioned for the first time; I do not find comments on this tool in the “Methods” section.

(A): Thank you very much. Further information has been added to the Methods section. Line 136

A complex case may be defined as a case scoring ≥100 points (total complexity), provided that the overall score for clinical management complexity items (CCI1 to CCI5) is ≥50 points.

(R): According to item CCI1 96,0 % of the patients were in need of palliative care. This seems not congruent to other figures in the “Results” section (see line 231) and should be addressed in the discussion.

(A): The item CCI1 of the Case Complexity Index distinguishes three levels

  • Level 1: 1 disabling condition with sudden onset (stroke, hip fracture)…)
  • Level 2: Has severe mental disorders or one severe advanced organic disease (grades III-IV on any scale)
  • Level 3: Needs palliative care (based on an ICD identification code or a specific scale or report)

No cases were reported for Level 1. For level 3 we have used the NECPAL CCOMS-ICO© 3.0 to identify patients with palliative care needs.

(R): CCI9: Taking into account the data in section 3.1, I would have expected that there should also have been some (few) patients without more severe impairment in the Barthel- and Pfeiffer-Score, respectively.

(A): 122 patients of our sample showed Dependency according to CCI9. It is indeed a high percentage. It is necessary to highlight that CCI9 combines dependency regarding basic activities of daily living (Barthel ≤ 15) and severe cognitive impairment (Pfeiffer ≥ 8 points). Nevertheless, it could be seen as contradictory with the data reported in section “3.1. Characteristics of the Sample”. This is because the percentages of dependency and cognitive impairment were reported from the VIG-Frailty scale that has different cut-off points, as it could be seen in Table 1. To avoid confusions in readers the following phrases have been changed from results section (Line 189).    

The mean Barthel Index value was 50.369 (SD=32.174). The average number of errors in Pfeiffer’s test was 4.9128

(R): CCI14: “Yes” in 86,6 % of the cases seems rather high.

(A): Thanks. “Yes” and “No” were mixed-up in CCI from CCI10 to CCI14. Some of them were reported and changed in the last review. It has been changed in Table 3. We have also reported that Item CCI11 was skipped from CCI table and has been added too.

(R): Line 218/219: “The levels of complexity defined by the IDC-Pal do not match the priority levels assigned by the nurses at the nursing homes.” These data are not presented in detail.

(A): Thanks. We have added a new column to Table 4 to present all the data for readers.
